# Investigating the Effect of Oxidants on the Quantification and Characterization of Charcoal in Two Southeast Australian Sedimentary Records

Mark Constantine IV [1],*, Xiaohong Zhu [1,2], Haidee Cadd [3,4] and Scott Mooney [1]

1 Earth and Sustainability Science Research Centre, University of New South Wales, Sydney 2033, Australia
2 School of Geography and Environment, Jiang Xi Normal University, Nanchang 330022, China
3 Chronos 14Carbon-Cycle Facility, University of New South Wales, Sydney 2033, Australia
4 School of Earth and Environmental Sciences, University of Wollongong, Wollongong 2522, Australia
* Correspondence: z5179298@unsw.edu.au

**Abstract:** This study examined the effects of commonly used oxidants in sedimentary macroscopic charcoal analysis on two sediment cores from Thirlmere Lakes National Park, Southeast Australia. The cores, from Lake Werri Berri (WB3) and Lake Couridjah (LC2), span ~900 years and 135,000 years, respectively. The Charcoal Accumulation Rate (CHAR) for both charcoal area and count was quantified using four different chemical treatments and compared to a control using only water. We also quantified the Charring Intensity (CI) of isolated charcoal fragments, a proxy for the severity/intensity of fire, determined using the FTIR spectral characteristics of the remaining charcoal after each treatment. We found significant differences in both the area and number of particles across all treatments in both cores. Significantly, we found substantial differences in CI between treatments, with few charcoal particles formed in low-severity fire (e.g., below ~400 °C or 3.0 °C.s.$10^6$) in groups treated with an oxidant. In contrast, the control group displayed a wider range of CI values and contained lightly pyrolyzed particles. This suggests that methods using an oxidant to concentrate sedimentary charcoal are potentially biasing records towards high-intensity or -severity fires. We suggest that consideration should be used when choosing laboratory methods based on the hypotheses being tested.

**Keywords:** charcoal analysis; wildfire; FTIR spectroscopy; Charring Intensity (CI); Australia; cultural burning

## 1. Introduction

The quantification of charcoal in sediments has been used as an indicator of past fires since the early work of Iversen [1]. The production of highly stable polyaromatics during combustion [2,3] and the common presence of charcoal in sediments [4] makes it a useful proxy for the identification or quantification of past fires [5,6]. Charcoal analysis has progressed from the quantification of microscopic (typically < 125 μm) charcoal alongside pollen in palynological studies (e.g., [7] to its own discipline with an overlapping group of methods. The recognition that 'macrocharcoal' (typically > 150 μm) more accurately reflects local (within ≈ $10^1$–$10^3$ m) fires [8], shifted attention to the quantification of larger charcoal particles (e.g., [8–10]) and the common use of wet sieving [11]. Many of these methods offer progress both in extracting the full range of charcoal fractions from sediment [12]. and in comparing oft-prescribed methods on parallel sediments [13].

As the development of more rigorous quantification methods has continued, there has been increased reliance on the use of light oxidants to concentrate charcoal from sediment (e.g., [14,15]). Although this can aid in the separation of charcoal from sediment, Constantine and Mooney [16] used laboratory-produced charcoal and commonly used image analysis software to demonstrate that it led to the potential loss of lightly pyrolyzed

material and, consequently, to a bias towards high-intensity fires being over-represented in charcoal records. The oxidative recalcitrance of charcoal is dependent on the arrangement of C-C bonds in its chemical structure. At higher temperatures, in both controlled conditions [17] and in wildfires [18], the chemical structure of charcoal forms into stable structures by decreasing the number of reactive sites on the charcoal.

In this research, we examined the effects of commonly used methods for concentrating charcoal from sediments on two cores from Thirlmere Lakes National Park, part of the Greater Blue Mountains World Heritage Area, located in eastern New South Wales, Australia. In particular, we were interested in determining if treatment choice affected both the quantity of charcoal present and the type of charcoal (e.g., charcoal formed under different pyrolysis conditions) from different-aged (recent to 130,000 years BP) sediments.

## 2. Methods

This study examines a range of commonly used charcoal analysis methods on two sediment cores and on laboratory-produced charcoal. The sediment cores, a 6.8 metre core from Lake Couridjah (code LC2) and a 22 cm monolith core from Lake Werri Berri (code WB3), were taken from Thirlmere Lakes National Park (34°13′ S, 150°32′ E) in south-eastern New South Wales, Australia. The age–depth model for the LC2 core was determined from a combination of radiocarbon and optically stimulated luminescence ages. Details of age determinations and age–depth modelling can be found in Forbes et al. [19] and Francke et al. [20]. The top 320 cm of the LC2 core encompasses part of Marine Isotope Stage 2—1 (present to 18,000 cal yr BP), and the bottom (320–680 cm) captures part of MIS 6-5 (105–135,000 years). The age–depth for WB3 (Figure S1) is a mixed model of $^{210}$Pb and radiocarbon ages created using rPlum [21,22]. Radiocarbon dating was conducted on pollen concentrates at the Chronos 14Carbon Cycle Facility, UNSW, following the procedures of Turney et al. [23] and Cadd et al. [24]. The deepest radiocarbon-dated sample from WB3 (20–21 cm) suggested a basal age of $916 \pm 12$ calibrated years before present (cal years BP). Both age–depth models are provided in the Supplementary Materials section.

A number of often-used charcoal analysis treatments (described below) along with a control (reverse osmosis $H_2O$) were carried out on sediment samples from both WB3 and LC2. WB3 was contiguously sampled in 1 cm intervals to test the effects of four treatments often used in charcoal analysis. Table 1 summarises the methods. Samples were categorised in terms of (1) strength of the chemical digestion method and (2) length of time to exposure, with T1 being the least reactive (no chemical exposure except $H_2O$) and T5 being the most chemically reactive treatment. Treatment 1 (T1) samples were treated as a control and were displaced in reverse-osmosis ($R_o$) water for 24 h. T2 samples were immersed in 2% bleach (sodium hypochlorite, NaOCl) for 24 h. T3 and T4 samples were immersed in a 4% solution of bleach for 6 h and 24 h, respectively. T5 used a treatment recently described for extracting and concentrating testate amoebae from minerogenic sediments [25], used here because it aims to deal with complex mixtures of organic and inorganic sediments. For T5, our samples were immersed in a 4% solution of sodium pyrophosphate ($Na_4P_2O_7$) and placed on a shaker table for 24 h in an effort to disperse the sediments. They were then wet sieved using a 250 μm sieve and the material retained on the sieve was washed into 20 mL of $R_o$ water, 25 mL of acetone (IUPAC name propan-2-one or $(CH_3)_2CO$), and 5 mL of NaOH and placed on a hotplate heated to 80 °C for 5 min. In this case, acetone was used as a co-solvent to increase the alkaline digestion of nonpyrolyzed organic matter.

Samples from the LC2 core were subsampled at 5–10 cm intervals and treated using a control (M1); immersion in reverse-osmosis ($R_o$) $H_2O$ for 24 h and a common preparation used in charcoal analysis (M2); and immersion in low-strength bleach (4% NaClO) for 24 h [26]. Sediment was measured and standardized volumetrically by measuring displacement when immersed in liquid (4% bleach or $H_2O$).

After treatment, the 250 μm size fraction of each treatment group (Table 1) was isolated by wet sieving [11] and hand-sorted in glass Petri dishes under a dissecting microscope. The charcoal fraction was quantified using the software program ImageJ version 1.52a [27]. The

concentration of charcoal in each treatment group was expressed as area ($mm^2/cm^3$) and count (no. of particles/$cm^3$). Charcoal area and count were then converted to a Charcoal Accumulation Rate (CHAR) by dividing charcoal concentration by the mean deposition time estimated from the age–depth model [11,28].

**Table 1.** Treatment methods tested on WB3 and Lc2 cores.

| Code | Chemical Treatment | Immersion Time |
| --- | --- | --- |
| **WB3** | | |
| **T1 (control)** | $H_2O$ ($R_o$) | 24 h |
| **T2** | 2% sodium hypochlorite (bleach) | 24 h |
| **T3** | 4% sodium hypochlorite (bleach) | 6 h |
| **T4** | 4% sodium hypochlorite (bleach) | 24 h |
| **T5** | sodium pyrophosphate/$H_2O$, $(CH_3)_2CO/NaOH$ | 24 h, 5 min at 80 °C |
| **LC2** | | |
| **M1 (control)** | $H_2O$ ($R_o$) | 24 h |
| **M2** | 4% sodium hypochlorite (bleach) | 24 h |

Charring Intensity (CI) was modelled using Fourier Transform Infrared (FTIR) Spectroscopy and multivariate statistics following the protocols of Constantine et al. [29]. CI, a proxy for fire severity/intensity, is a measure of exposure to heating (e.g., temperature and time of exposure). Charcoal produced under high energy (as measured in $kW/m^2$, e.g., heat release rate, total heat release, and duration) loses oxygenated functional groups of cellulose and lignin, resulting in stable, well-organized, graphite-like materials (dominated by aromatic carbon = carbon structures). Charcoal formed at a lower heat release rate, lower total heat release, or for lower duration is complex, disorganized, and has a greater proportion of aliphatic organic compounds that are susceptible to oxidation [30]. These chemical differences can be used to infer pyrolysis conditions (e.g., wildfire intensity) under which charcoal was formed [29]. From each sampled depth and treatment group where charcoal was present, a total of 1–8 macroscopic charcoal particles (the size fraction greater than 250 μm) were scanned using a PerkinsElmer ATF-FTIR spectrometer. The spectra were measured for wavenumbers from 4000 to 650 $cm^{-1}$ at 4 $cm^{-1}$ resolution with a wavenumber spacing of 1, with 4 scans accumulated per sample. The R package 'baseline' version 1.2-1 [31,32] was used to baseline-correct the spectra using the Modified Polynomial Fit function [33] CI was subsequently measured on individual charcoal particles for which FTIR spectra were measured (WB3 core, $n$ = 91 for T1, $n$ = 76 for T2, $n$ = 88 for T3, $n$ = 78 for T4, and $n$ = 78 for T5. LC2 core, $n$ = 345 for M1, $n$ = 495 for M2). The modelled minimum, maximum, and mean values of each treatment group were examined to determine which treatment(s) captured the largest range (minimum and maximum CI values) to best describe changes in fire severity/intensity in both cores.

## 3. Results

### 3.1. WB3

The CHAR and CI results of each treatment group are displayed in Figure 1 and Table 2 (Figures S2 and S3 display results by depth). There was a strong, positive correlation between particle count (no./$cm^3$) and area ($mm^2/cm^3$) within treatment groups (Table 2, *Spearman's Correlational Coefficient*), suggesting both measurements faithfully captured similar aspects of fire events. However, there were significant differences between treatments at the same sample depths. Treatment groups T1-4 displayed similar median charcoal count values, though their minimum and maximum values differed. A one-way ANOVA (Table 3) was performed to compare the effect treatments had on count and area and revealed significant differences between groups in both particle count (($F$) 4 = [9.349], $p$ = $1.56^{-6}$) and area (($F$) 4 = [9.932], $p$ = $7.43^{-7}$). Table 2 summarises the mean values and standard deviations of each group. T4 had the greatest average number of particles while T3 had the highest charcoal area.

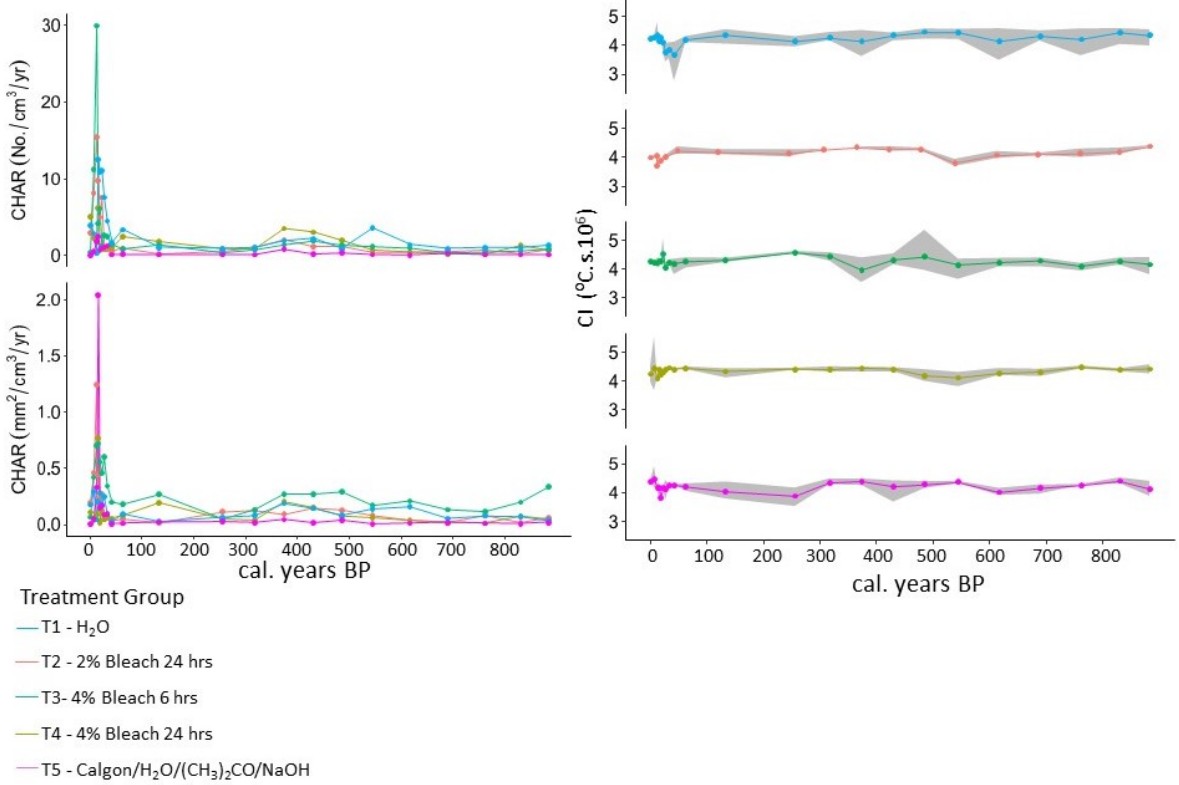

**Figure 1.** The results of the five treatment groups. CHAR count (No./cm$^3$/yr), CHAR area (mm$^2$/cm$^3$/yr), and CI. Treatment groups are represented by colour. The grey shading in CI represents the minimum and maximum CI values.

**Table 2.** Group average charcoal count (no./cm$^3$), area (mm$^2$/cm$^3$), and mean size (mm$^2$) of charcoal particles from treatment groups from the WB3 core. The Spearman range correlation coefficient was calculated between charcoal count and area for each treatment.

| Treatment Group | Count No./cm$^3$ μ | Count No./cm$^3$ σ | Area mm$^2$/cm$^3$ μ | Area mm$^2$/cm$^3$ σ | Spearman Rank Correlation |
|---|---|---|---|---|---|
| **T1—H$_2$O (Control)** | 35.50 | 21.32 | 3.40 | 3.57 | 0.873 |
| **T2—2% NaClO** | 45.80 | 41.38 | 2.64 | 2.54 | 0.897 |
| **T3—4% NaClO** | 43.82 | 26.32 | 7.31 | 4.79 | 0.583 |
| **T4—4% NaClO** | 64.48 | 41.13 | 3.59 | 3.16 | 0.783 |
| **T5—Calgon/$H_2O$, $(CH_3)2CO/NaOH$** | 9.77 | 7.76 | 1.22 | 1.35 | 0.690 |

**Table 3.** One-way ANOVA between groups results for charcoal concentration (as no./cm$^3$ and mm$^2$/cm$^3$) and mean particle size of charcoal (mm$^2$).

| Group | SS | df | MS | F | *p*-Value |
|---|---|---|---|---|---|
| **Charcoal (no./cm$^3$)** | 34,872 | 4 | 8718 | 9.349 | $1.53^{-6}$ *** |
| **Charcoal (mm$^2$/cm$^3$)** | 448.7 | 4 | 112.17 | 9.932 | $7.43^{-7}$ *** |

The significance of *p*-values '***' 0.001.

A one-way ANOVA found significant differences in CI between treatment groups ((F) 4 = [6.330], *p* = 0.000133). The mean CI values were similar across treatment groups but there were significant differences in the range (Figure S3). T1 had the highest range (2.1) while T2 had the lowest (0.8). T4 and T3 both had higher maximum values than other

treatment groups (5.5 and 5.4, respectively) while T1 had significantly lower minimum CI values than all other groups. A Tukey's Honestly Significant Test (HSD) (Figure S4 and Table S1) revealed a significant pair-wise difference in mean CI between T2-T1, T3-T2, T4-T2, and T5-T2.

*3.2. LC2*

Because of the large sedimentary hiatus in the LC2 core [19], CHAR and CI were considered over both the entire core length and separately (S2; MIS 6-5 and S1; MIS 2-1). Tables 4 and 5 and Figure 2 *(Figure S5 provides results by depth)* display the CI and CHAR (area ($mm^3/cm^3$/yr) and number of particles (No./$cm^3$/yr)) results of LC2. Significantly fewer subsamples had no charcoal in the M1 (2/95) than M2 (26/102) treatment groups. Paired sample t-tests suggest significant differences between both count: (M1) ($\mu$ = [297.05], $\sigma$ = [448.12]) and (M2) ($\mu$ = [25.71], $\sigma$ = [41.66]); t(90) = [1.98668], *p* = [$8.07226^{-8}$] and area: (M1) ($\mu$ = [50.16], $\sigma$ = [124.81]) and (M2) ($\mu$ = [11.51], $\sigma$ = [29.46]); t(90) = [0.003056], *p* = [1.9867].

**Table 4.** Summary statistics of Charring Intensity (CI mean, minimum, maximum, and range) and charcoal concentration (count and area) across the two preparation methods: M1 using water and M2 using dilute bleach.

| Treatment Group | CI (Mean) | CI (Min) | CI (Max) | CI (Range) | Area ($mm^2/cm^3$) | Count (No./$cm^3$) |
|---|---|---|---|---|---|---|
| **M1** | 3.8 | 3.1 | 4.4 | 1.3 | 48.85 | 289.57 |
| **M2** | 4.1 | 3.4 | 5.1 | 1.7 | 10.60 | 24.35 |

**Table 5.** Summary of average values of M1 and M2 treatment groups in the upper core (S1) and lower core (S2). Minimum, maximum, and range of CI and charcoal influx area ($mm^2/cm^3$) and count (No./$cm^3$) across sections S1 and S2 are displayed.

| Treatment Group/Depth | CI (Mean) | CI (Min) | CI (Max) | CI (Range) | Area ($mm^2/cm^3$) | Count (No./cm) |
|---|---|---|---|---|---|---|
| **M1-S1** | 3.2 | 2.3 | 3.8 | 1.5 | 32.78 | 232.29 |
| **M1-S2** | 4.3 | 3.8 | 4.8 | 1.0 | 61.05 | 333.06 |
| **M2-S1** | 4.1 | 3.3 | 5.0 | 1.7 | 8.86 | 17.67 |
| **M2-S2** | 4.1 | 3.5 | 5.2 | 1.7 | 11.93 | 29.41 |

There was a marked increase in mean CI between S1 (3.2) and S2 (4.3) in M2. The average range in S1 (1.5) was higher than S2 (1.0). The average minimum CI (2.3) and maximum CI (3.8) values of S1 were also lower than S2 (min = 3.8, max = 4.8). In the M2 treatment, between 170 and 285 cm, there was a significant decrease in CI values compared to the rest of the core. The mean CI in M1 was more consistent throughout the length of the record than it was in M2. The mean CI was 4.1 for S1 and 4.2 for S2 for the M1 treatment. The average minimum CI for S1 was 3.3 and 3.5 for S2, while the average maximum was 5.0 for S1 and 5.3 for S2. The range of CI values was the same (1.7) in both S1 and S2.

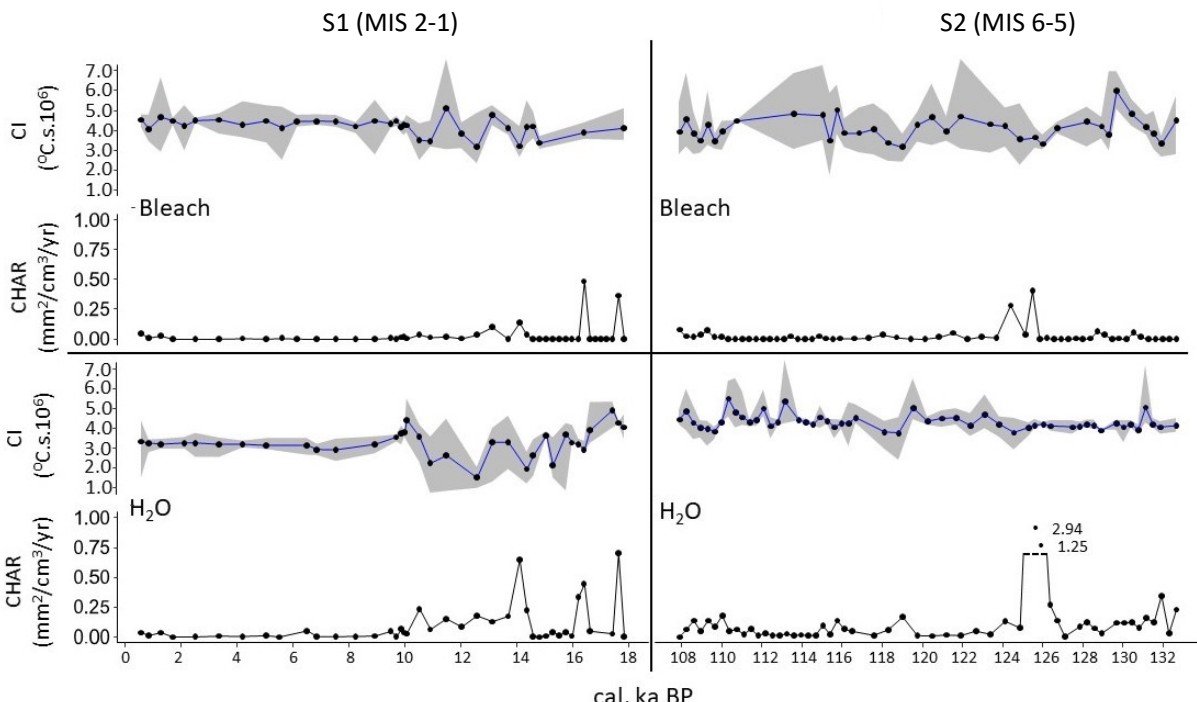

**Figure 2.** Comparison of charcoal area ($mm^2/cm^3/yr$) and Charring Intensity (CI) results of the LC2 core following Bleach (4% NaOCl) and $H_2O$ treatments. CI is presented as the mean (blue line) and minimum and maximum values (grey shading). Charcoal values were truncated at 1 $mm^2/cm^3$ in the $H_2O$ treatment group to emphasize depths with lower accumulation values. The true values at ~125–126 cal. ka BP are marked. See Supplementary Information for CI and charcoal accumulation by depth.

## 4. Discussion

Prior to these analyses, it was hypothesized that the use of oxidant treatments would result in lower charcoal particle and area amounts and the loss of lightly pyrolyzed material. In the WB3 core, there was a strong positive correlation (Table 4) between charcoal count and area in all treatment groups, suggesting that they were all faithfully capturing similar aspects of the fire regime. However, there were some unexpected results. For example, there was good agreement in charcoal peaks between treatment groups but, in some cases, there were offsets of 1–2 cm between groups. Edwards and Whittington [34], Schlachter and Horn [35], and, more recently, Tsakiridou et al. [13] have argued that variation between sediments sampled from the same catchment could be the result of spatial heterogeneity between cores. Tsakiridou et al. [13], for example, used horizontally adjacent sediments and found general consistency across treatments, but noted that their subsamples displayed charcoal peaks at different depths in some instances, which they suggested could be due to uneven surface features in the bog at the time of deposition. The WB3 monolith is wider than a regular core, and the site has been a shallow wetland with aquatic macrophytes growing across it in the past, which might explain the discrepancies or offsets between the samples.

Despite advances in extracting charcoal from sediment, less attention has been placed on the chemical dissimilarities between charcoal formed at different temperatures, a useful metric in assessing the severity/intensity of past wildfires [29,36,37]. Constantine and Mooney [16] demonstrated that light oxidants degrade charred material formed below ~450 °C (~3.1 °C.s.$10^6$), potentially eliminating charcoal formed in low-intensity fires from the sedimentary record. These types of low-intensity fires are those often described as used by indigenous peoples to modify vegetation and for other resource management and cultural practices [38]. This is particularly relevant to studies in Australia, where it has been

suggested that Aboriginal people often intensively managed landscapes with 'fire-stick farming' [39], a fire regime that is often characterised as high frequency, small in spatial scale, and 'cool' (low severity/intensity). It is commonly argued that this systematic utilisation of fire in the landscape (as opposed to fire in a domestic setting) was used to facilitate hunting, for resource manipulation (e.g., for gathering food stuffs or other resources that might be stimulated or advantaged by fire), for 'cleaning-up' (or socialising sensu [40]) the landscape, to facilitate movement, or to reduce hazards in fire-prone ecosystems [41]. In areas where low-intensity fires are a part of the fire regime, and in particular ones where a low-intensity fire regime is purposefully maintained, care should be taken to ensure that charcoal formed in low-severity fires is not eliminated [42].

Syntheses of charcoal records are frequently used to examine biomass burning over regional and larger spatial scales (e.g., [43,44]) often in conjunction with pollen or other proxy records (e.g., [39,45]) to infer not only changes in fires but also to identify humans as a primary ignition source. However, much of this research may only be capturing a portion of the record, one biased towards high-intensity fires. Implicit in this bias is the loss of the low-intensity signal, likely reducing the detection of low-intensity, frequent patch fires purported to be used by indigenous people. The under detection of these types of fire regimes suggests careful consideration should be given when using these records to discuss low-intensity, anthropogenic fire regimes.

In the LC2 core, the greater amount of charcoal recovered in M1 can be ascribed to the fraction of highly reactive charcoal not preferentially removed by bleaching [16]. The control (M1) treatment resulted in a greater number of charcoal particles and fewer sediment subsamples with no recorded charcoal particles, suggesting a more detailed and unbiased [16] record of fire is available when no oxidant is used. There were also substantial differences in CI values between the treatment groups, particularly in sediments from the MIS 2-1 period of the core (S1). Interestingly, sections with high charcoal concentration did not coincide with increased CI in the M1 or M2 treatments, which suggests the relationship between large fires that consume large amounts of biomass (high CHAR) and intense/severe fires that produce high temperatures (high CI) is not straightforward.

The CI of WB3 showed more subtle differences between treatment groups. Similar to the CHAR values, the mean CI of the five WB3 treatment groups was broadly consistent, highlighting that the CI metric is replicable in adjacent sediments. However, there were differences in both the range of CI measured and its upper and lower values. The lack of charcoal measured below CI 3.0 °C.s.$10^6$ in T2-5 CI (the minimum CI was 3.6 °C.s.$10^6$) in all treatment groups, compared to the water control (T1) (minimum CI 2.8 °C.s.$10^6$), indicated that oxidants preferentially removed lightly pyrolyzed charcoal and, with it, evidence of low-severity/intensity fires. This was the case in both the LC2 and WB3 records. In order to fully encapsulate past fire intensity using the charring intensity metric, care should be taken in choosing an extraction method that does not bias the fire record toward high-severity/intensity fires.

## 5. Conclusions

Two sediment cores from Southeast Australia, one spanning ~1000 years and the other ~135,000 years, were examined using a number of commonly prescribed methods for separating and concentrating charcoal from sediments. We found that treatment choice affects the amount of charcoal recovered from sediments, particularly in sedimentary deposits comprising lightly pyrolyzed charcoal. The removal of lightly pyrolyzed material potentially results in fire records biased towards high-severity fires. This has implications for the examination of frequent, small-patch, anthropogenic ignitions such as the type often ascribed to Aboriginal people in Australia. Particularly when evaluating CI in a sediment core, and especially at sites where people are thought to have modified the 'natural' fire regime through the use of frequent, low-intensity fires, it may be beneficial to forgo oxidants in order to retain charcoal covering the full range of potential charring intensities. It may also be beneficial to test different treatments on a subset of sediment samples from the

core being examined to find a treatment that minimizes the destruction of lightly charred material while still retaining some of the benefits associated with using a light oxidant to concentrate charcoal.

**Supplementary Materials:** The following supporting information can be downloaded at: https://www.mdpi.com/article/10.3390/fire6020054/s1, Figure S1: ABayesian age-depth model for Lake Werri Berri (WB3) using radiocarbon and lead-210 dates. The age depth model was created using the mixed modelling R program 'rplum' [21]. Figure S2: The results of the five treatment groups on WB3. (A) charcoal particle count (No./$cm^3$), (C) charcoal particle area ($mm^2$/$cm^3$). Box and Whisker plot of mean (B) charcoal count, (D) charcoal area. The box represents the median, 25th and 75th quartile values. The whiskers represent the minimum and maximum values. Outliers are represented by red dots. Figure S3: (A) Charring Intensity ($°C.s.10^6$) of WB3 treatment groups. On the left-hand side (A) mean CI is presented as a black line and dots. Maximum and minimum values are presented as the grey shaded area. In (B) the range of Charring Intensity is presented for the five treatment groups. Figure S4: Tukey's HSD test of significance of mean Charring Intensity for each treatment group of WB3. Group comparisons that deviate to the left or right of 0.0 indicate significant differences between groups. This revealed that T2-T1, T3-T2, T4-T2, and T5-T2 showed significant differences. Figure S5: The results of the two treatment groups on LC2. Charcoal concentration is presented as area ($mm^2$/$cm^3$) and count (No./$cm^3$). Charring Intensity is presented as a mean (black line) and minimum and maximum values (grey shading). In the $H_2O$ treatment group, values at ~550 cm are truncated at 200 (area) and 1500 (count), respectively, to highlight variation in depths with lower charcoal concentration. Table S1: Results of Tukey's HSD test of significance on mean Charring Intensity for WB3.

**Author Contributions:** Conceptualization, M.C.IV and S.M.; methodology, M.C.IV and S.M.; formal analysis, M.C.IV; investigation, M.C.IV and X.Z.; writing—original draft preparation, M.C.IV; writing—review and editing, S.M. and H.C.; visualization, M.C.IV. All authors have read and agreed to the published version of the manuscript.

**Funding:** This project was partially funded by Australian Research Council (ARC) grant DP200101123 'Shaping a sunburnt country: fire, climate and the Australian landscape' to Tony Dosseto, Scott Mooney, Ross Bradstock, Damien Lemarchand, Nathalie Vigier. Mark Constantine IV's research is partially supported by an Australian Government Research Training Program (RTP) Stipend and RTP Fee-Offset Scholarship through Federation University Australia.

**Data Availability Statement:** Please contact corresponding author for data.

**Acknowledgments:** Special thanks to Tim Cohen and Matt Forbes for their generous help and access to the LC2 core.

**Conflicts of Interest:** The authors declare no conflict of interest.

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
