# Peer review of "Investigating the Effect of Oxidants on the Quantification and Characterization of Charcoal in Two Southeast Australian Sedimentary Records"

_fire, doi:10.3390/fire6020054_

Round 1
Reviewer 1 Report
- Page 7, Line 1-6: Please provide more specific why this work is interesting, what actually brings something new and what results it presents.
- Please expand the sentence: "if treatment choice affected bot the quantity of charcoal present and the type of charcoal from different aged sediments".The results showed or give answer.....
- Page 6 Line 5. Please describe more precisely section, 3.1WB3 is to short, and next sections also
- Page 6 Line 7. There is a strong, positive correlation between.... Can you make short discussion why?
- Please correct some editorial errors, missing parentheses, etc.
- Please try to reorganize the article, it is very hard to read and understand.
I'm not sure what is the most important aspect of this work.
Reviewer 2 Report
In this paper, effects of various oxidation treatment on detection sensitivity and accuracy of fire records were investigated. The introduction and discussion sections are well organized. In my view, this work can be accepted after answering the following questions:
(1) Formats mistakes should be avoided, such as ml and 14C.
(2) Information of WB3 and LC 2 had better be added in Table 1 for ease understanding of sample treatments.
(3) Structural analysis and data processing methods should be described specifically, especially FTIR and calculation of CI.
(4) How can fire severity be evaluated by CI? What does “Big fire” and “hot fire” mean in Line 20 of Page 12?
(5) As emphasized in Abstract and Conclusion sections, utilization of various oxidants resulted in elimination of lightly-pyrolyzed charcoal particles. In my view, influencing mechanism of oxidants had better be offered based on approximate structural analysis, for better understanding of formation of bias.
